# Chemical bonding concepts emerge naturally from maximally entangled atomic orbitals

Lexin Ding[1,2], Eduard Matito[3,4,5] & Christian Schilling [1,2] ✉

Chemical bonding is a nonlocal phenomenon that binds atoms into molecules. Its ubiquitous presence in chemistry, however, stands in stark contrast to its ambiguous definition and the lack of a universal perspective for its understanding. In this work, we rationalize and characterize chemical bonding through the lens of an equally nonlocal concept from quantum information, the orbital entanglement. We introduce the maximally entangled atomic orbitals (MEAOs) whose entanglement pattern is shown to recover both Lewis (two-center) and beyond-Lewis (multicenter) structures, with multipartite entanglement serving as a comprehensive index of bond strength. Our unifying framework for bonding analyses is effective not only for equilibrium geometries but also for transition states in chemical reactions and complex phenomena such as aromaticity. It also has the potential to elevate the Hilbert space atomic partitioning to match the prevalent real-space partitioning in the theory of atoms in molecules. Accordingly, our work provides a new framework for understanding fuzzy chemical concepts using rigorous, quantitative descriptors from quantum information.

The chemical bond is a central concept in chemistry, originating from Gilbert N. Lewis's idea of electron pairs shared between atoms[1]. Lewis's theory, developed independently of quantum mechanics, is remarkably simple and profoundly influential. It is thus commonly taught already at the high school level. Heitler and London provided the first quantum mechanical description of a hydrogen molecule bond[2], later evolving into valence bond theory[3,4], which uses superpositions of configurations of electrons occupying atomic orbitals. Molecular orbital theory[5] followed historically, combining atomic into molecular orbitals before occupying them with electrons. Both approaches link bonding to quantum mechanics by virtue of rudimentary wave functions, guided by chemical intuition, and together they form the foundation of modern bonding theories.

Contrarily, modern electronic structure methods based on sophisticated wave function ansätze offer great accuracy[6–10] but at the same time complicate intuitive bonding descriptions. To address this issue, various chemical bonding tools have emerged[11–15], categorized into real-space or topological, and Hilbert space approaches. Concerning the former category, approaches such as Bader's quantum theory of atoms in molecules[16] and the electron localization function[17,18] rely on topological analyses of the electron or same-spin pair density. Consequently, they require computationally intensive integrations over three-dimensional regions to derive descriptors like bond order and aromaticity indices[19,20]. While being largely basis-set independent, these methods therefore suffer from scalability issues and integration errors[21]. Due to these issues, a recent information theoretical approach to bonding analysis based on real-space partitioning, though conceptually interesting, has thus far been limited to mean-field descriptions[22]. In contrast, Hilbert space methods[11,23–25] partition molecular systems into atomic domains using

[1]Faculty of Physics, Arnold Sommerfeld Centre for Theoretical Physics (ASC), München, Germany. [2]Munich Center for Quantum Science and Technology (MCQST), München, Germany. [3]Donostia International Physics Center (DIPC), Donostia, Spain. [4]Ikerbasque Foundation for Science, Bilbao, Spain. [5]Donostia Polimero eta Material Aurreratuak: Fisika, Kimika eta Teknologia, Kimika Fakultatea, Euskal Herriko Unibertsitatea (EHU), Donostia, Spain. ✉e-mail: c.schilling@physik.uni-muenchen.de

basis functions, offering computational efficiency but facing challenges in basis set dependence and ambiguity in the orbital assignments[26]. Recent orbital localization schemes, such as intrinsic atomic orbitals (IAOs)[25], mitigate these limitations, providing a robust foundation for chemical bonding analyses, yet without inherently yielding bonding concepts.

Despite these challenges, Hilbert space partitioning methods hold significant promise. As we will demonstrate, these methods are particularly well-suited for applying tools from quantum information theory (QIT) to analyze chemical bonding, offering a perspective that addresses their current limitations. To further motivate our work, we recall that recent studies[27–37] have applied QIT tools to partitions based on molecular orbitals, thus quantifying electron correlation and the representational complexity of molecular quantum states. In contrast, quantifying bonding structures with an emphasis on bonding orders is a significantly more challenging task, as it requires adapting the QIT approach to the nonlocal nature of covalent bonds by instead referring to fully localized orbitals. The primary accomplishment of our work is to devise a general scheme for computing maximally entangled atomic orbitals (MEAOs), whose entanglement patterns quantitatively recover both Lewis (two-center) and beyond-Lewis (multicenter) bonding structures, with multipartite entanglement serving as a robust index of bond strength.

Accordingly, by leveraging parallels between chemical bonding and quantum entanglement, our work provides a QIT-based framework that captures crucial features of chemical bonding such as hybridization, bond orders, multicenter bonding, conjugation, and aromaticity without a priori chemical assumptions. Our framework, which we validate on standard and unconventional bonding cases, thus provides a new framework for understanding fuzzy chemical concepts using rigorous, quantitative descriptors from quantum information theory.

## Results

### Entanglement in a single covalent bond

We motivate our general QIT-based framework by first analyzing the idealized covalent bond formed by two electrons in a symmetric diatomic molecule. For this, let $\varphi_L$ and $\varphi_R$ be the two relevant (real-valued) atomic orbitals localized on the left and right atomic centers, with an overlap $S = \langle \varphi_L | \varphi_R \rangle \neq 0$. The prototypical bonding state $|\Psi_{bond}\rangle$ is then obtained by occupying the corresponding bonding orbital with the electron pair,

$$|\Psi_{bond}\rangle = |\uparrow\downarrow\rangle_\phi \otimes |0\rangle_{\bar{\phi}}, \tag{1}$$

where $\phi$ and $\bar{\phi}$ denote the bonding and antibonding orbitals,

$$\phi \equiv \frac{\varphi_L + \varphi_R}{\sqrt{2(1+S)}}, \quad \bar{\phi} \equiv \frac{\varphi_L - \varphi_R}{\sqrt{2(1-S)}}. \tag{2}$$

Here and in the following, we use the formalism of 'second quantization' and recall that any orbital $\varphi$ gives rise to a four-dimensional Fock space, spanned by the states $|0\rangle_\varphi, |\uparrow\rangle_\varphi, |\downarrow\rangle_\varphi$, and $|\uparrow\downarrow\rangle_\varphi$, where the arrow denotes the electron spin and $|0\rangle_\varphi$ the vacuum state. It is crucial to note that the idealized state, Eq. (1), takes the form of a product state (in 2nd quantization) and, accordingly, it does not contain any entanglement between the bonding and antibonding orbitals. This is by no means a contradiction to the common expectation: Entanglement is a relative concept and depends on the division of the total system into orbital subsystems. By referring to molecular orbitals $\phi, \bar{\phi}$, the corresponding orbital entanglement merely quantifies the validity of the independent electron-pair approximation rather than the bonding order. This, in turn, reflects very well the conceptually different perspectives of molecular orbital and valence bond theory.

According to the definition of effective bond order (EBO)[15], which is the difference between the occupation numbers of the bonding and antibonding orbitals divided by 2, the state in Eq. (1) represents a "perfect" single bond with EBO = 1. In general, to compute the EBO for more realistic molecular wave functions, one would first need to categorize various molecular orbitals as bonding, antibonding, or nonbonding. This can be based on factors such as spatial symmetry, or their ability to promote or inhibit electron sharing between the atomic centers[11,38]. However, such criteria can become ambiguous and arbitrary in multicenter molecules, potentially leading to erroneous results. As our work will show, it is precisely the QIT framework used in a valence bond theoretical context that offers excellent prospects for overcoming these deficiencies of approaches based on molecular orbital theory.

To elaborate further on the idealized covalent bond and align our QIT perspective with its nonlocal character, we introduce the symmetrically orthogonalized atomic orbitals

$$\widetilde{\varphi}_{L/R} = \frac{1}{2}\left(\frac{1}{\sqrt{1+S}} + \frac{1}{\sqrt{1-S}}\right)\varphi_{L/R} + \frac{1}{2}\left(\frac{1}{\sqrt{1+S}} - \frac{1}{\sqrt{1-S}}\right)\varphi_{R/L}. \tag{3}$$

In this orbital basis, the state in Eq. (1) has the following form:

$$|\Psi_{bond}\rangle = \frac{1}{2}\Big(|0\rangle_L \otimes |\uparrow\downarrow\rangle_R + |\uparrow\rangle_L \otimes |\downarrow\rangle_R - |\downarrow\rangle_L \otimes |\uparrow\rangle_R + |\uparrow\downarrow\rangle_L \otimes |0\rangle_R\Big). \tag{4}$$

That is, it is maximally entangled relative to the orbitals $\widetilde{\varphi}_L$ and $\widetilde{\varphi}_R$, with an entanglement value $E = \log(4)$. Here, we used the fact that the entanglement $E$ for pure states follows as the von Neumann entropy $S(\hat{\rho}) \equiv -\text{Tr}[\hat{\rho}\log(\hat{\rho})]$[39],

$$E(|\Psi_{bond}\rangle\langle\Psi_{bond}|) = S(\hat{\rho}_{L/R}), \tag{5}$$

of the corresponding orbital reduced density matrices (RDM)

$$\hat{\rho}_{L/R} = \text{Tr}_{R/L}[|\Psi_{bond}\rangle\langle\Psi_{bond}|]. \tag{6}$$

We also recall that the maximal entanglement between two subsystems of dimension $d$ (in our case $d = 4$) is simply $E_{max} = \log(d)$. This entanglement value indicates maximal correlation between the physical observables measured on the left and right orbital, which, by construction, recovers crucial bonding features such as electron sharing and spin pairing.

Finally, we would like to stress that the orbitals $\widetilde{\varphi}_L$ and $\widetilde{\varphi}_R$ define a partition of the underlying one-particle Hilbert space, which conceptually resembles the real-space partitioning in the quantum theory of atoms in molecules[16]. In more general situations involving multiple atomic centers, or when each atomic center hosts more than one atomic orbital, calculating the entanglement between any two atomic subspaces becomes significantly more challenging for several reasons. First, the reduced state $\hat{\rho}_{AB}$, defined on two atomic centers $A$ and $B$, is high-dimensional because the Fock spaces of $A$ and $B$ grow exponentially with the number of orbitals on each center. Second, $\hat{\rho}_{AB}$ is generally a mixed state as a result of the coupling of $A$ and $B$ to other atomic centers. Consequently, the closed formula Eq. (5) for entanglement in pure states is no longer applicable, and the entanglement between $A$ and $B$ must instead be determined numerically as the minimum "distance" from $\hat{\rho}_{AB}$ to the set of

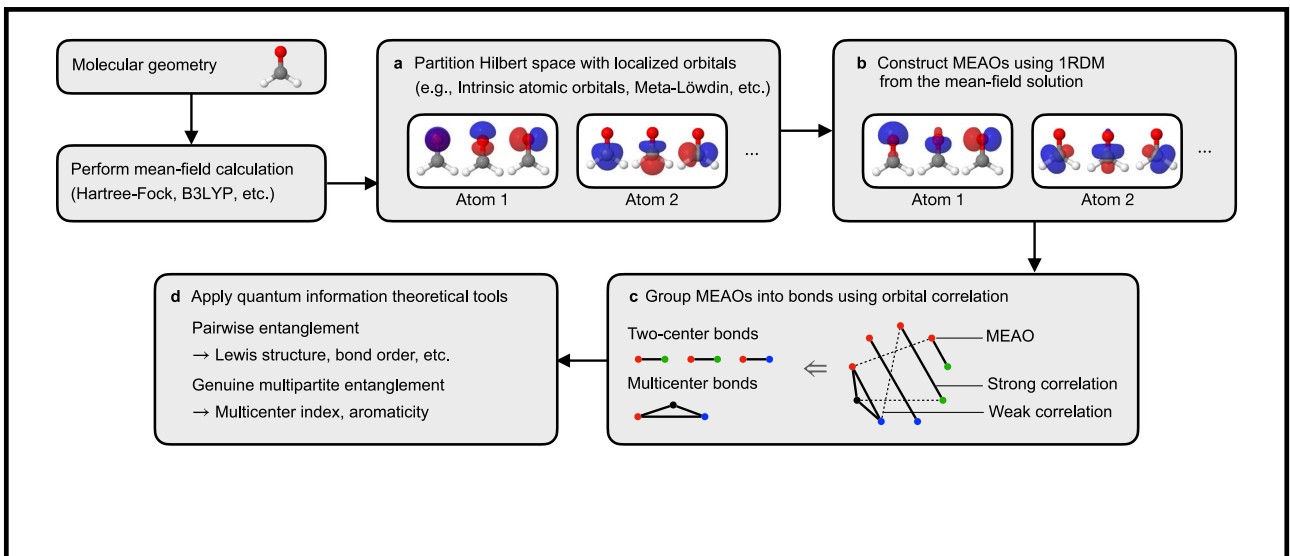

**Fig. 1 | An overview of bonding analysis using maximally entangled atomic orbitals (MEAOs). a** Perform a Hilbert space partition with localized orbitals. Each group of localized orbitals defines the local Hilbert space for each atom. **b** Construct MEAOs via internal orbital rotations that preserve the local Hilbert spaces, and minimize the objective function, Eq. (14), computed with a mean-field wave function. **c** Group MEAOs into two-center and multicenter bonds by identifying strong links in the orbital correlation graph. MEAOs are represented by the vertices in the correlation graph whose colors indicates the atomic assignment, whereas strong and weak orbital correlations are illustrated with the solid and dashed edges, respectively. **d** Apply quantum information theoretical tools to each identified bond.

unentangled mixed states $S_{\text{sep}}$[40,41]

$$E(\hat{\rho}_{AB}) = \min_{\hat{\sigma} \in S_{\text{sep}}} \text{Tr}[\hat{\rho}_{AB}(\log \hat{\rho}_{AB} - \log \hat{\sigma})],$$
$$S_{\text{sep}} = \left\{ \sum_i p_i \hat{\sigma}_A^{(i)} \otimes \hat{\sigma}_B^{(i)}, p_i > 0, \sum_i p_i = 1 \right\}. \quad (7)$$

Nevertheless, the high dimensionality of $\hat{\rho}_{AB}$ can often be circumvented when the entanglement between $A$ and $B$ is predominantly captured by a few pairs of localized orbitals $\hat{\rho}_{ij}$. This is a crucial aspect of our work which we will elaborate on in the next section.

## Maximally entangled atomic orbitals

The centerpiece of our results is the set of maximally entangled atomic orbitals (MEAOs). We briefly outline their construction and usage here, while detailed procedures are provided in the Methods section. Given a Hilbert space partition $\Pi = \{\mathcal{H}^{(1)}, \mathcal{H}^{(2)}, \ldots, H^{(M)}\}$ with atomic subspaces $\mathcal{H}^{(m)}$, MEAOs are a basis of localized orbitals $\mathcal{B}^{\text{MEAO}} = \{\{\phi_i^{(1)}\}, \{\phi_j^{(2)}\}, \ldots, \{\phi_k^{(M)}\}\}$ that ideally maximizes the total inter-center entanglement, defined as the sum of entanglement between two orbitals sitting on different atoms

$$E_{\text{inter-center}} = \sum_{i<j} E(\hat{\rho}_{ij}), \quad (8)$$

where $\hat{\rho}_{ij}$ is the orbital RDM of orbital $i$ and orbital $j$ that sits on different atomic centers, obtained by tracing out all other orbital degrees of freedom from the overall wave function $|\Psi\rangle$. While Eq. (8) is physically well motivated, its direct evaluation is computationally demanding due to the mixed-state nature of $\hat{\rho}_{ij}$ and its dependence on the four-particle RDM. To identify the MEAOs, we need a computationally affordable objective function to guide the orbital rotations. To this end, we observe that the entanglement in a two-orbital RDM $\hat{\rho}_{ij}$ is maximized when it exactly matches the single covalent bond state, Eq. (4). The entanglement in the state, Eq. (4), can be attributed to two important superpositions: (i) $|0\rangle_i \otimes |\uparrow\downarrow\rangle_j$ and $|\uparrow\downarrow\rangle_i \otimes |0\rangle_j$; (ii) $|\uparrow\rangle_i \otimes |\downarrow\rangle_j$ and $|\downarrow\rangle_i \otimes |\uparrow\rangle_j$. These two superpositions are purely two-body, and the coherence therein is measured by the elements $\Gamma_{j\uparrow,j\downarrow}^{i\uparrow,i\downarrow}$ and $\Gamma_{j\uparrow,i\downarrow}^{i\uparrow,j\downarrow}$,

respectively, of the two-particle reduced density matrix (2RDM) $\Gamma_{k\sigma,l\tau}^{i\sigma,j\tau} = \langle\Psi|f_{i\sigma}^{\dagger}f_{j\tau}^{\dagger}f_{l\tau}f_{k\sigma}|\Psi\rangle$. Based on this observation, we define the objective function for obtaining the MEAOs as

$$F_{\text{MEAO}}(\mathcal{B}) = \sum_{i<j} \left|\Gamma_{j\uparrow,j\downarrow}^{i\uparrow,i\downarrow}\right|^2 + \left|\Gamma_{j\uparrow,i\downarrow}^{i\uparrow,j\downarrow}\right|^2, \quad (9)$$

where orbitals $i$ and $j$ belong to different atomic centers, and

$$\mathcal{B}^{\text{MEAO}} = \arg\min_{\mathcal{B} \sim \Pi} F_{\text{MEAO}}(\mathcal{B}), \quad (10)$$

where the minimization is performed over all bases of localized orbitals that respect the Hilbert space partition $\Pi$ (denoted as $\mathcal{B} \sim \Pi$). In Fig. 1, we summarize the workflow of using MEAO for bonding pattern analysis:

a. Choose a Hilbert space partitioning.
b. Construct MEAOs using a mean field wave function $|\Psi^{\text{MF}}\rangle$ and Eqs. (14), (10).
c. Compute the mutual information $I_{ij} = S(\hat{\rho}_i) + S(\hat{\rho}_j) - S(\hat{\rho}_{ij})$ for all pairs of inter-center orbitals $i$ and $j$, and apply graph-theoretical methods to group the MEAOs into bonding pairs or bonding groups.
d. Evaluate the pairwise or multipartite entanglement (defined in section "Beyond-Lewis structures: genuine multipartite entanglement") for these MEAO groups, using a correlated wave function $|\Psi\rangle$.

## Lewis structures and bipartite entanglement

Recovering standard chemical concepts: Within the framework of standard Lewis theory, molecular bonding is described by Lewis structures, where bonds are represented by pairs of shared electrons depicted as lines between atoms. The bond multiplicity is then given by the number of lines connecting two atoms. What sets MEAOs apart from other localized orbitals is their ability to naturally reflect the Lewis structures of molecules through their correlation and entanglement properties.

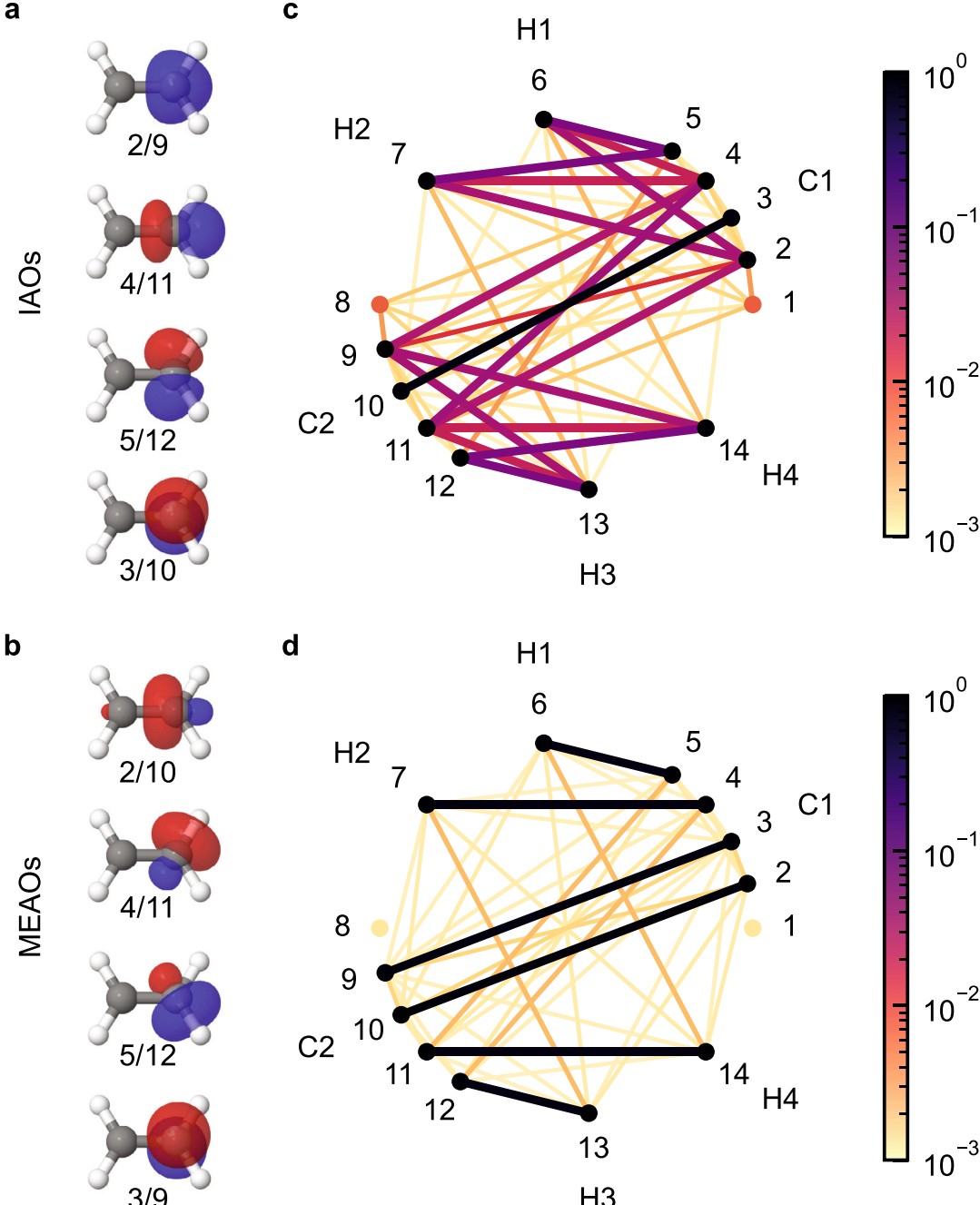

**Fig. 2 | Orbital isosurfaces and orbital correlation graphs of intrinsic atomic orbitals (IAOs) and maximally entangled atomic orbitals (MEAOs) in the ground state of C₂H₄. a** Isosurfaces of valence carbon IAOs. **b** Isosurfaces of valence carbon MEAOs. **c** Orbital correlation graph of IAOs. Values of the normalized single orbital entropy $S(\hat{\rho}_i)/\log(4)$ and the normalized orbital-orbital correlation $I_{ij}/\log(16)$ are represented in log-scale by the color of the nodes and edges of the graph, respectively. **d** Same as **c** but for MEAOs. The calculations are performed with the cc-pVDZ basis.

In Fig. 2, we compare the shapes and orbital correlations of the valence IAOs and MEAOs of ethene (C₂H₄) with the cc-pVDZ basis. The MEAOs are constructed from the 14 minimal IAOs using a Hartree-Fock solution. Subsequently, two complete active space (CAS) calculations are performed using the IAOs and MEAOs, respectively, to compute orbital correlation and entanglement. While the IAOs retain much of their free atomic character, with a clear distinction between one 2*s* and three 2*p* orbitals, the MEAOs exhibit the classical *sp*² hybridization of carbon, forming three *sp*² orbitals and one 2*p* orbital. This remarkable emergence of hybridization—a fundamental concept in organic chemistry—demonstrates how MEAOs naturally capture chemical intuition (see Supplementary Note 2 for further examples).

The differences in orbital shapes between IAOs and MEAOs result in distinct correlation graphs, where MEAOs provide a sparse and interpretable picture of bonding. High pairwise correlation values ($I_{ij} \approx I_{max}$) clearly identify bonding interactions, with non-bonding pairs showing negligible correlation values ($I_{ij} \sim 10^{-3} I_{max}$). For ethene, the correlation graph of the MEAOs reveals six bonding interactions: four C–H σ bonds and two C-C bonds (one σ and one π), as shown in Table 1. In contrast, the correlation graph of the IAOs is dense and does not provide a clear bonding picture.

**Table 1 | Pairwise correlation $I_{ij}$ and entanglement $E_{ij}$ between the bonding maximally entangled atomic orbitals (MEAOs) in the ground states of various molecules**

|  | Bond | $I_{ij}/I_{max}$ | $E_{ij}/E_{max}$ |
|---|---|---|---|
| $CH_4$ | C–H | 0.957 | 0.954 |
| $C_2H_6$ | C–H | 0.928 | 0.929 |
|  | C–C | 0.955 | 0.946 |
| $C_2H_4$ | C–H | 0.914 | 0.917 |
|  | C–C($\sigma$) | 0.944 | 0.932 |
|  | C–C($\pi$) | 0.918 | 0.895 |
| $C_2H_2$ | C–H | 0.906 | 0.910 |
|  | C–C($\sigma$) | 0.915 | 0.908 |
|  | C–C($\pi_1$) | 0.915 | 0.875 |
|  | C–C($\pi_2$) | 0.915 | 0.875 |
| $N_2$ | N–N($\sigma$) | 0.962 | 0.950 |
|  | N–N($\pi_1$) | 0.919 | 0.880 |
|  | N–N($\pi_2$) | 0.919 | 0.880 |
| $Li_2$ | Li–Li | 0.942 | 0.944 |
| LiH | Li–H | 0.767 | 0.768 |
| LiF | Li–F | 0.191 | 0.192 |
| $He_2$ | He–He | 0 | 0 |

The correlation and entanglement values are normalized by their maximally attainable values $I_{max} = 2\log(4)$ and $E_{max} = \log(4)$, respectively. All calculations are performed using a complete active space including all electrons in the minimal intrinsic atomic orbital subspace, with the experimental equilibrium geometry and with the cc-pVDZ basis.

For each two-center bond, there exists a pair of MEAOs with correlation ($I_{ij}/I_{max}$) and entanglement ($E_{ij}/E_{max}$) values close to 1. The number of such pairs matches the bond orders of the molecules analyzed in Table 1. Furthermore, deviations of $I_{ij}/I_{max}$ or $E_{ij}/E_{max}$ from 1 reflect deviations of the ground state from the idealized Lewis structure[42]. The results also highlight chemical trends, such as that $\pi$ bonds being generally lower in covalency than $\sigma$-bonds. In summary, MEAOs reorganize the ground state wave function into a representation as closely aligned as possible with the Lewis structure of the molecule, providing a robust framework for quantifying bond orders through orbital-orbital entanglement.

Before we move on to more advanced applications of the MEAO formalism, we first point out that orbital entanglement directly captures only bonding effects arising from electron sharing and spin coupling, namely covalent bonding. Ionic or intermolecular bonding effects are only indirectly shown in the reduction of orbital entanglement. To demonstrate this constraint, we studied three molecules with increasing levels of ionicity, $Li_2$, LiH, and LiF. As shown in the second panel in Table 1, we indeed observe a consistent decrease going from $Li_2$ to LiF, in both mutual information and entanglement between the two most correlated MEAOs. Finally, we test against the van der Waals molecule $He_2$, which was found to have zero correlation and entanglement between the MEAOs on the two helium atoms. We remark that this number of absolute zero is ensured by the fact that the minimal IAO subspace from which the He-MEAOs are constructed consists of only one orbital for each atom. And since both electrons are placed on this orbital, the total ground state is simply a product state between the two fully occupied He-MEAOs. To fully confirm the lack of covalency in $He_2$, we extended the minimal IAO subspace to include the full basis set, only to find a vanishingly small amount of correlation between the two atomic centers.

The challenging harpoon mechanism: While the singlet ground state of LiH at equilibrium ($R_{Li\text{-}H} \approx 1.6$ Å) is predominantly ionic, it transitions to a covalent bonding character of the singlet first excited state at an avoided crossing around $R_{Li\text{-}H} = 3$ Å (see Fig. 3a). This

transition is accompanied by a shift of electron density from the hydrogen atom towards the center of the molecule. Therefore, when the molecule is stretched to dissociation, the electron sharing first increases as it enters the covalent phase, and then decreases due to final dissociation. This process is described as the harpoon mechanism[43]. The hallmark of this mechanism is a peak in the covalent bond order around the avoided crossing[44], which is confirmed by the electron delocalization index $\delta_{Li\text{-}H}$[20,42] (Fig. 3b). This index measures the covariance of electron populations in the Li and H atoms within a real-space partition (see Supplementary Note 4 for a definition). The bonding in this system is particularly challenging to describe because standard Mulliken-like electron-sharing analyses based on Hilbert space partitioning fail to detect the signature peak in the bond order[45]. By contrast, the quantum information approach using MEAOs successfully identifies this challenging feature, underscoring the strength of our framework in capturing complex bonding phenomena.

In Fig. 3, we present the low-lying energy spectrum $\{\mathcal{E}_i\}$ of LiH calculated using the aug-cc-pVDZ basis set, the electron delocalization index $\delta_{Li\text{-}H}$ of the ground state (taken from ref. 44), and the highest entanglement between a Li-MEAO and a H-MEAO in the approximate thermal state. This thermal state includes the lowest four eigenstates $|\Psi_i\rangle$, given by

$$\widehat{\rho}(\beta) = \frac{1}{Z(\beta)} \sum_{i=0}^{3} \exp(-\beta\mathcal{E}_i)|\Psi_i\rangle\langle\Psi_i|, \tag{11}$$

where $\beta = 10^3$ Ha$^{-1}$ represents the inverse temperature. Incorporating a low-temperature thermal state is necessary because the energy gap between the singlet ground state and the threefold degenerate triplet first excited level practically closes around $R_{Li\text{-}H} = 4$Å, rendering the ground state alone an insufficient physical representation of the molecule. At lower separations ($R_{Li\text{-}H} < 4$ Å), the thermal state is effectively dominated by the ground state, as the gap remains nonzero. An extended analysis of entanglement in both the ground and excited states is provided in Supplementary Note 3.

Around equilibrium, there is still a considerable amount of entanglement between a Li-MEAO and a H-MEAO in the thermal state $\widehat{\rho}(\beta)$, which closely resembles the ground state at this geometry. A fully ionic state would correspond to a product state with zero entanglement relative to the atomic partition. However, since the state retains some covalent character, the entanglement remains finite (see Supplementary Note 3 for the relationship between $E_{Li\text{-}H}$ and the ionic character of the bond). Remarkably, the entanglement $E_{Li\text{-}H}$ increases as the molecule dissociates, peaking around $R_{Li\text{-}H} = 3$ Å, coinciding with the avoided crossing. Beyond this peak, the entanglement decays to zero as dissociation progresses. This behavior stands in stark contrast to the monotonic decrease of bond order observed in Mulliken-like analyses[45], demonstrating how the MEAO formalism effectively captures subtle and complex bonding phenomena, such as the harpoon mechanism. These results underscore the strength of MEAOs as a comprehensive framework for analyzing challenging and nontrivial bonding scenarios.

## Beyond Lewis structures: genuine multipartite entanglement

Genuine multipartite entanglement: While the majority of molecules conform to the Lewis paradigm of two-electron, two-center bonds, many molecules cannot be fully described by this model. These molecules often exhibit bonding structures involving more than two atomic centers. For two-center bonds, the connection between bonding and entanglement is established through the bipartite maximally entangled state $|\Psi_{bond}\rangle$ in Eq. (4). The natural question arises: what is the equivalent of $|\Psi_{bond}\rangle$ in the case of multicenter bonds?

To motivate our approach to the multicenter bonding problem, we first highlight that multipartite entangled states can belong to different classes with distinct internal structures. For example, for three

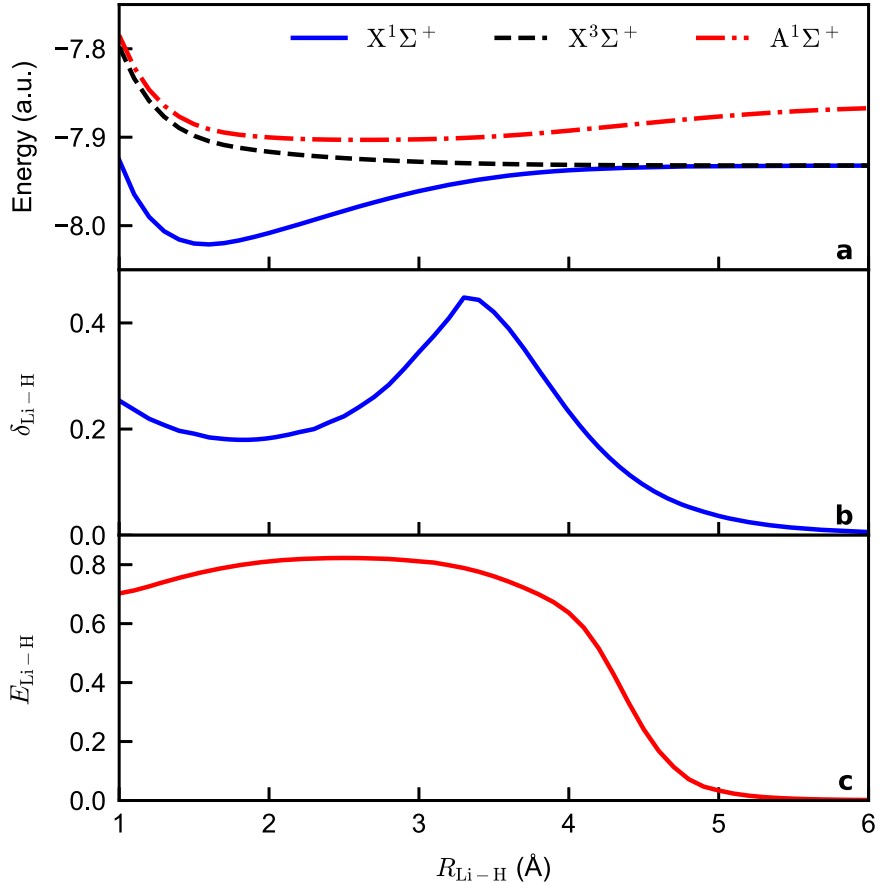

**Fig. 3 | LiH dissociation in the aug-cc-pVDZ basis. a** Energies (in atomic units, a.u.) of the lowest two states in the spin singlet sector ($X^1\Sigma^+$ and $A^1\Sigma^+$) and the triply degenerate first excited state in spin triplet sector ($X^3\Sigma^+$). **b** The electron delocalization index $\delta_{Li-H}$ of the singlet ground state. **c** The highest entanglement value (normalized by log(4)) between two maximally entangled atomic orbitals, one localized on Li and and one on H, in the thermal state at $\beta = 10^3\,\mathrm{Ha}^{-1}$ involving the singlet ground state and the triply degenerate triplet first excited states.

qubits, the following two states[46]:

$$|\text{GHZ}\rangle = \frac{1}{\sqrt{2}}(|000\rangle + |111\rangle),$$

$$|\text{W}\rangle = \frac{1}{\sqrt{3}}(|100\rangle + |010\rangle + |001\rangle),$$

(12)

belong to separate tripartite entanglement classes and are both maximally entangled within their respective classes. Notably, while every pair of qubits in $|\text{W}\rangle$ is entangled, no qubit pairs are entangled in $|\text{GHZ}\rangle$. Instead, the entanglement in $|\text{GHZ}\rangle$ exists collectively among all three qubits.

A $K$-partite pure state $|\Psi\rangle$ is said to exhibit genuine multipartite entanglement (GME) if it cannot be expressed as a product state under any bipartition[47]. Otherwise, the state is called biseparable. To determine whether $|\Psi\rangle$ contains GME, one can check whether the entropy of any subsystem is nonzero. Based on this principle, a measure for pure state GME is defined as[48,49]:

$$\text{GME}(|\Psi\rangle) = \min_A S(\widehat{\rho}_A),$$

(13)

where $A$ runs over all possible subsystems, which, in our case, correspond to collections of orbitals. When $K = 2$, GME reduces to the standard bipartite entanglement measure Eq. (5) for pure states. The GME measure Eq. (13) is normalized by its maximal value $\log(d)$, where $d$ is the dimension of the Fock space of the smallest subsystem. This measure can also be extended to mixed states via the convex roof construction[50–52]. For example, the states $|\text{GHZ}\rangle$ and $|\text{W}\rangle$ have distinct GME values of $\log(2)$ and $\log(3) - \frac{2}{3}\log(2)$, respectively, demonstrating the capacity of GME to quantify entanglement genuinely shared among multiple orbitals, regardless of the specific multicenter bonding type or internal entanglement structure.

To illustrate the utility of GME as a multicenter bonding index, we analyze prototypical three-center, two- and four-electron bonds in the ethyl cation ($C_2H_5^+$) and the allyl anion ($C_3H_5^-$), respectively. In both molecules, a three-orbital cluster is detected from the MEAO correlation graph based on a B3LYP calculation, and the cluster is treated with a CAS. The remaining two-orbital, two-electron bonds are treated with mean-field accuracy, which ensures the correct electron count in the CAS. The MEAO framework thereby enables the automatic detection of electron-deficient and hypervalent bonds. As shown in Table 2, the constructed CAS directly identifies the three-center, two-electron bond in $C_2H_5^+$ and the three-center, four-electron bond in $C_3H_5^-$. The multicenter bonding character of these molecules are confirmed by the high GME values (0.891 for $C_2H_5^+$ and 0.787 for $C_3H_5^-$, both normalized by log(4)), as are all subsequent mentions of explicit values of GME. To further validate the universal applicability of GME as a multicenter bond index beyond carbon rings, we also analyzed lithium clusters. While lithium dimers are described by standard two-center two-electron bonds, lithium trimer anions ($Li_3^+$) and tetramers $Li_4$ display clear multicenter bonding patterns, with GME value 0.910 and 0.904, respectively.

GME as an aromaticity index: Aromaticity is a subtle chemical concept[53], manifested in many chemical and physical properties[54,55],

**Table 2 | Genuine multipartite entanglement (GME) in the ground state of various molecules**

| | CAS(#o, #e) | GME / log(4) |
|---|---|---|
| $C_2H_5^+$ | (3, 2) | 0.891 |
| $C_3H_5^-$ | (3, 4) | 0.787 |
| $Li_3^+$ | (3, 2) | 0.910 |
| $Li_4$ | (4, 4) | 0.904 |
| $C_6H_6$ | (6, 6) | 0.967 |
| $C_5H_5N$ | (6, 6) | 0.953 |
| $1, 2\text{-}C_4H_4N_2$ | (6, 6) | 0.950 |
| $1, 3\text{-}C_4H_4N_2$ | (6, 6) | 0.952 |
| $1, 4\text{-}C_4H_4N_2$ | (6, 6) | 0.955 |
| $1, 3, 5\text{-}C_3H_3N_3$ | (6, 6) | 0.952 |
| $C_5H_5^-$ | (5, 6) | 0.954 |
| $C_4H_5N$ | (5, 6) | 0.725 |
| $C_4H_4O$ | (5, 6) | 0.572 |
| $C_5H_6$ | (4, 4) | 0.343 |
| $C_6H_{12}$ | (6, 6) | 0.015 |
| $C_6H_{10}$ | (6, 6) | 0.035 |
| $C_6H_8$ | (6, 6) | 0.039 |

The values of GME are normalized by the maximally attainable value log(4). The computation of the GME for each molecule uses the solution of a complete active space (CAS) consisting of the orbitals in the largest inseparable cluster of size #o in the orbital correlation diagram and correlating #e electrons. The orbital correlation diagrams are obtained using a B3LYP calculation with the 6-311++G(d,p) basis set.

but its underlying origin lies in the highly delocalized nature of electrons within a ring[56]. This delocalization, a hallmark of aromatic systems, is naturally captured by the GME.

The most prominent example of an aromatic molecule is benzene, where the six out-of-plane $p$-orbitals collectively form a highly delocalized six-center bond, reflected by a six-$\pi$-orbital cluster with a GME value of 0.970 with the experimental geometry[57] and the cc-pVDZ basis set. The mutual information between the MEAOs of benzene is presented in Fig. 4, showing a distinct six-orbital cluster formed by six $\pi$-MEAOs. A complete active space configuration interaction calculation involving six electrons in six orbitals was performed for this cluster, while all other orbitals were treated at the Hartree-Fock level. In contrast to benzene, the highest six-orbital GME values for three six-member rings − cyclohexane (no $\pi$ bond), cyclohexene (one $\pi$ bond), and cyclohexa-1,3-diene (two $\pi$ bonds) − are 0.015, 0.035, and 0.039, respectively. These results confirm that $n$-center electron delocalization is a necessary condition for an $n$-partite GME value to approach one. To benchmark GME as a quantitative aromaticity index, we selected challenging systems from a collection of aromaticity tests[58].

First, we analyze nitrogen-substituted benzene rings. Replacing one or more carbon atoms in the benzene ring with nitrogen disrupts the uniform electron delocalization, reducing aromaticity[59]. This reduction is reflected in the GME values listed in Table 2, where the aromaticity of substituted species is slightly lower than that of benzene. The exact ordering of aromaticity for these six-membered rings differs depending on the aromaticity index used, such as the multicenter index (MCI)[60], harmonic oscillator measure of aromaticity (HOMA)[61,62], aromatic fluctuation index (FLU)[63], and

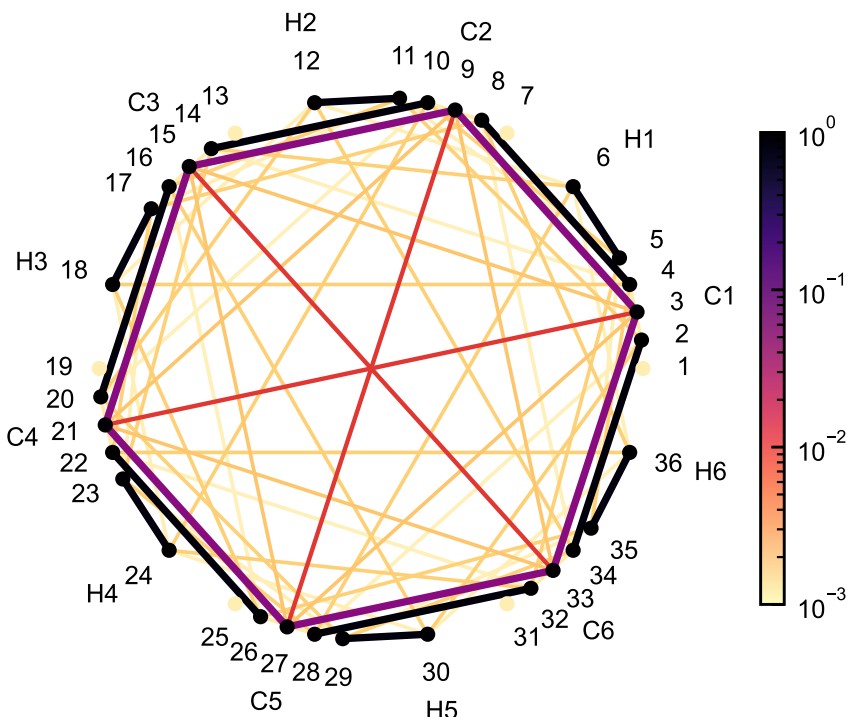

**Fig. 4 | Orbital correlation graph of $C_6H_6$.** Values of the normalized single orbital entropy $S(\hat{\rho}_i)/\log(4)$ and the normalized orbital-orbital correlation $I_{ij}/\log(16)$ are represented in log-scale by the color of the nodes and edges of the graph, respectively. The calculation is performed with the experimental equilibrium geometry and with the cc-pVDZ basis.

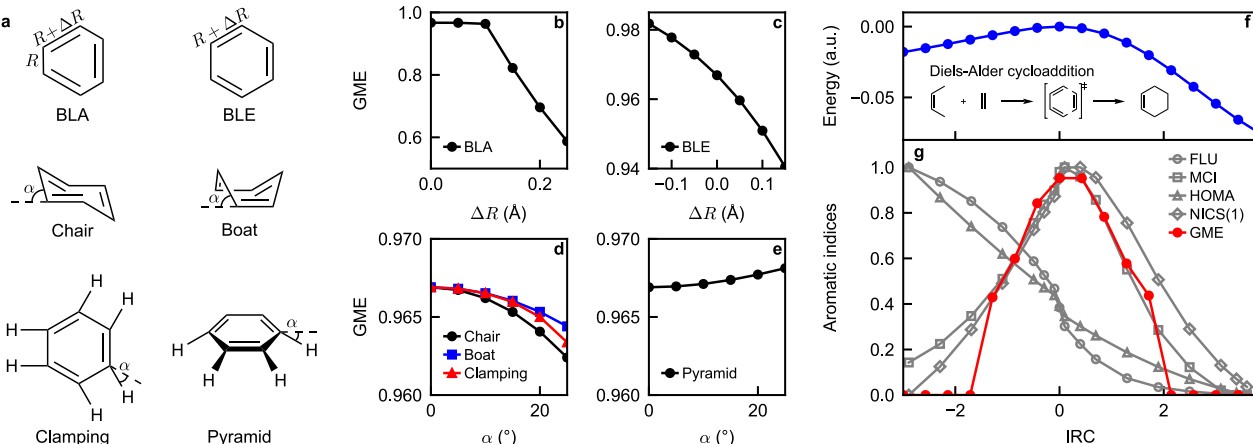

**Fig. 5 | Tests of genuine multipartite entanglement (GME) as an aromatic index.** **a** Six types of deformations of the benzene ring, including bond length alternation (BLA), bond length elongation (BLE), chair, boat, clamping, and pyramid. **b** GME as a function of the displacement $\Delta R$ in the BLA test. **c** GME as a function of the displacement $\Delta R$ in the BLE test. **d** GME as a function the deformation angle $\alpha$ in the chair (black circle), boat (blue square), and clamping (red triangle) test. **e** GME as a function of the deformation angle $\alpha$ in the pyramid test. **f** Energy (subtracted by the transition state energy, in atomic units, a.u.) of the ground state of the reaction systems along the intrinsic reaction coordinates (IRC) of a Diels-Alder cycloaddition. **g** Aromatic fluctuation index (FLU, grey circle), multicenter index (MCI, grey square), harmonic oscillator measure of aromaticity (HOMA, grey triangle), nucleus-independent chemical shift (NICS(1), grey diamond), and GME (red circle filled) of the ground state of the reaction systems along the IRC. Aromatic indices are linearly transformed to the interval [0, 1] for visibility. GME values are normalized by log(4).

nucleus-independent chemical shift (NICS)[64] (see Supplementary Note 4 for definitions). Notably, some well-known indices, such as HOMA and the para-delocalization index, failed this test[58]. The GME values, however, correctly capture the trend that benzene has the highest aromaticity compared to nitrogen-substituted rings.

Second, we examine five-member rings, which can host five-orbital six-electron singlet bonds. A canonical example is $C_5H_5^-$, whose aromaticity is lower than benzene due to reduced point-group symmetry and the lack of particle-hole symmetry in the $\pi$-subspace. As expected, the GME value for $C_5H_5^-$ is lower than that for benzene. Substituting one carbon atom with nitrogen reduces the GME to 0.725, while substituting with oxygen reduces it further to 0.572. For comparison, the non-aromatic $C_5H_6$ yields a low GME value, and the detected cluster contains only four orbitals. These results confirm that GME accurately captures different levels of aromaticity in five-member rings.

Third, we investigate the effect of geometric deformation on the GME of benzene. The benzene ring is highly stable due to the resonance energy associated with its aromatic structure. Consequently, deviations from its equilibrium geometry are expected to raise the ground state energy and reduce aromaticity[65]. We analyze six types of deformations: (1) bond length alternation (BLA), where C-C bond lengths alternate with a difference of $\Delta R$; (2) bond length elongation (BLE) by $\Delta R$; (3) clamping; (4) boat; (5) chair; and (6) pyramid (see Fig. 5a for graphical depictions). The GME values under these deformations are presented in Fig. 5b-e.

The GME is most sensitive to BLA, which explicitly breaks the six-fold symmetry of the C-C bond lengths. For BLE, we find that the GME decreases monotonically as the C-C distance increases, due to reduced overlap between the $\pi$-orbitals. However, the reduction in GME under BLE is less pronounced compared to BLA. Other deformations also reduce GME, as expected, with the exception of the pyramid deformation, where the GME slightly increases. This anomaly likely arises because the pyramid deformation preserves the six-fold symmetry of the ring. Overall, GME agrees well with the expectation that deviations from the equilibrium geometry reduce aromaticity, reinforcing its validity as a quantitative aromaticity index.

Aromaticity in transition states: During chemical reactions, electrons can become highly delocalized, sometimes forming transient aromatic ring structures. A concrete example is the Diels-Alder cycloaddition[66,67], where a conjugated butadiene reacts with ethene to form cyclohexene. Although neither the reactants nor the product is aromatic, the six $\pi$-electrons involved are temporarily shared across the entire ring as the $\pi$-bonds in butadiene and ethene break to form two new $\sigma$-bonds and a new $\pi$-bond, thereby promoting aromaticity[68,69] (see Fig. 5f for a graphical description of the reaction).

In Fig. 5f,g, we present the ground state energy of the system, calculated using the hybrid functional B3LYP[70] with the 6-311++G(d,p) basis set along the reaction pathway parameterized by the intrinsic reaction coordinate (IRC). We also include the GME values of the ground states based on corresponding active space calculations, alongside several aromaticity indices (shifted and renormalized to the interval [0, 1]; details in Methods). The multicenter bonding clusters and active spaces are determined using the graph partitioning technique described in Methods. Negative and positive IRC values indicate directions toward the reactants and products, respectively.

The Diels-Alder reaction presents a stringent test for aromaticity indices, as some widely used measures, such as HOMA[61,62] and FLU[63], fail to detect aromaticity in the transition state[58]. Remarkably, GME exhibits a distinctive peak corresponding to a six-orbital cluster, with the location of the peak aligning well with other successful indices[58]. Data points where GME = 0 correspond to regions where no prominent six-member clusters are detected. The successful detection of aromaticity in the transition state underscores the robustness of GME as an aromaticity index, even for systems far from equilibrium.

In summary, we have introduced GME as an index for multicenter bonding, particularly aromaticity, motivated by its high value in benzene. Through a series of tests, we confirmed that GME accurately assesses aromaticity across various aromatic molecules, including substituted and deformed benzene rings, while recovering well-known chemical trends both at equilibrium and in transition states. What sets GME apart is its conceptual foundation, rooted in the same formalism that characterizes two-center bonds[71]. Practically, the MEAO framework enables fully automatic identification of multicenter bonding clusters and their intensities, requiring no manual intervention. This

**Table 3 | Pairwise correlation $I_{ij}$ and entanglement $E_{ij}$ between the bonding MEAOs in the ground states of CO and Cr(CO)$_6$**

|  | Bond | $I_{ij}/I_{max}$ | $E_{ij}/E_{max}$ |
|---|---|---|---|
| CO | C-O($\sigma$) | 0.980 | 0.981 |
|  | C-O($\pi_1$) | 0.872 | 0.873 |
|  | C-O($\pi_2$) | 0.872 | 0.873 |
| Cr(CO)$_6$ | Cr-C | 0.684 | 0.721 |
|  | C-O($\sigma$) | 0.905 | 0.919 |
|  | C-O($\pi_1$) | 0.627 | 0.665 |
|  | C-O($\pi_2$) | 0.627 | 0.665 |

The correlation and entanglement valuesc are normalized by their maximally attainable values $I_{max} = 2\log(4)$ and $E_{max} = \log(4)$, respectively.

makes GME a powerful tool for exploring complex bonding scenarios with minimal effort.

### Bonding in transition metal complexes

Our final demonstration of the MEAO framework involves the chromium hexacarbonyl, Cr(CO)$_6$, a prototypical transition-metal complex. As an isolated molecule, CO exhibits a characteristic triple bond, comprising one $\sigma$ and two $\pi$ components. Upon coordination of the six CO ligands to the Cr center, the C-O bond order decreases to a value between a double and a triple bond, in accordance with the well-known Dewar-Chatt-Duncanson mechanism of $\sigma$-donation and $\pi$-back-donation[72,73]. This reduction in covalency is also quantitatively shown by the diminished electron sharing between the C and O atoms in a subsequent study[74]. In Table 3, we present the correlation and entanglement between the bonding MEAOs in both the free carbon monoxide and the chromium hexacarbonyl, correctly confirming the reduction in the covalent bond order between C and O. The MEAOs are optimized based on an IAO Hilbert space partition constructed from the entire atomic basis set (6-31G(d,p) basis set for carbon and oxygen, and Roos augmented double-zeta atomic natural orbital basis for chromium). Correlation and entanglement between the MEAOs are evaluated from the one-particle reduced density matrix (1RDM) obtained in a B3LYP density-functional-theory calculation.

### Discussions

Turning the fuzzy concept of chemical bonding into quantitative descriptors is essential for advancing our understanding of molecular properties and reaction mechanisms. In this work, we introduced an automated framework for versatile bonding analysis, combining the intuitive principles of valence bond theory with the rigorous tools of quantum information. At the core of this framework are the maximally entangled atomic orbitals (MEAOs), whose entanglement patterns naturally recover both Lewis (two-center) and beyond-Lewis (multi-center) bonding structures across a wide range of molecules. Moreover, our approach successfully captures challenging bonding scenarios that elude some widely adopted bonding descriptors, including the harpoon mechanism in LiH dissociation and aromaticity in the transition state of a Diels-Alder reaction.

Our framework offers a transformative perspective on the nature of chemical bonding. Established concepts such as electron sharing and spin pairing are seamlessly integrated into the entanglement between (or among) spatially localized orbitals. The chemical significance of MEAOs, which are automatically hybridized and intuitively meaningful, underscores a deep connection between the quantum mechanical essence of bonding and the entanglement between atomic subspaces. Furthermore, by leveraging insights from entanglement theory, our framework provides a unified characterization of two-center and multicenter bonds using the same tools. Notably, the LiH example demonstrates that the less commonly used Hilbert space atomic partition can be just as effective for bonding analysis as real-

space partitioning, provided the correct descriptors, orbital entanglement in this case, are employed. This finding suggests a paradigm shift, where Hilbert space approaches could emerge as strong alternatives to real-space methods, offering advantages such as scalability and reduced sensitivity to integration errors.

Our work suggests several promising directions for future research. First, applying the MEAO formalism to monitor bond-breaking and bond-formation processes in molecular dynamics would be of practical interest. Second, while this study focused primarily on aromaticity as a multicenter bonding example, the efficacy of genuine multipartite entanglement in detecting other multicenter bonds, such as those in boranes[75], agostic bonds[76], and hydrogen bonds[77], warrants further exploration. Lastly, extending the treatment of multicenter bonds beyond a single active space description represents a compelling challenge. This could be addressed from a quantum information perspective, through the development of advanced multipartite entanglement measures for mixed states, or from a quantum chemistry perspective, by designing wave function methods capable of handling multiple active spaces simultaneously, while representing each multicenter bond by a pure state.

In conclusion, our framework bridges the gap between intuitive chemical bonding theories and quantitative quantum descriptors, offering a unified, scalable, and insightful approach for analyzing a wide array of bonding scenarios. By integrating concepts from quantum chemistry and quantum information theory, this work sets the foundation for a deeper understanding of chemical bonding and its role in molecular behavior.

## Method

Here we present a detailed account of the construction and usage of MEAOs for bonding analysis.

Step 1: Hilbert space partitioning. To introduce the MEAOs, we begin with an atomic partition $\Pi = \{\mathcal{H}^{(m)}\}_{m=1}^{M}$ of the one-particle Hilbert space $\mathcal{H} = \bigoplus_{m=1}^{M} \mathcal{H}^{(m)}$, where $M$ is the number of atoms in the molecule. Typically, any procedure for such Hilbert space partitioning identifies both the subspaces $\mathcal{H}^{(m)}$ corresponding to individual atoms $m$ and the localized orbitals $\{\phi_i^{(m)}\}$ that form an orthonormal basis for $\mathcal{H}^{(m)}$. Common examples of these localized orbitals include IAOs[25] and Meta-Löwdin localized orbitals[78]. In principle, each atomic subspace $\mathcal{H}^{(m)}$ can be spanned by infinitely many possible orbital bases $\mathcal{B}^{(m)} = \{\phi_i^{(m)}\}$, with any two such bases being related by an orbital rotation that preserves the subspace $\mathcal{H}^{(m)}$. When we perform orbital optimizations in later steps, we always consider orbital rotations that preserves the Hilbert space partition $\Pi$ only.

Step 2: Constructing MEAOs. Guided by our analysis of the idealized covalent bond in Section "Entanglement in a single covalent bond", the central idea of our approach is to identify the distinctive orbital basis $\mathcal{B}_{MEAO}$ that maximizes the sum of inter-center orbital entanglements, $\sum_{i<j} E(\widehat{\rho}_{ij})$, where the summation include orbital pairs $(i, j)$ belonging to different atoms only. While direct computation of the entanglement $E(\widehat{\rho}_{ij})$ defined in Eq. (7) for all orbital pairs in larger systems can become computationally expensive, this approach offers valuable insight into the nature of chemical bonding by prioritizing the most significant inter-center correlations. Specifically, in the single covalent bond state, Eq. (4), both the particle numbers and the spin of the orthogonal atomic orbitals are perfectly correlated, as reflected in the maximal values of the coherent terms $\langle 0, \uparrow\downarrow | \widehat{\rho}_{ij} | \uparrow\downarrow, 0 \rangle$ and $\langle \uparrow, \downarrow | \widehat{\rho}_{ij} | \downarrow, \uparrow \rangle$, where $\widehat{\rho}_{ij} = |\Psi_{bond}\rangle\langle\Psi_{bond}|$ (see Supplementary Note 1 for a more detailed analysis). Inspired by this observation, we construct a proxy objective function for orbital entanglement that preserves the key features of the MEAOs in the perfect bonding state $|\Psi_{bond}\rangle$:

$$F_{MEAO}(\mathcal{B}) = \sum_{i<j} \left|\Gamma_{j\uparrow,j\downarrow}^{i\uparrow,i\downarrow}\right|^2 + \left|\Gamma_{j\uparrow,i\downarrow}^{i\uparrow,j\downarrow}\right|^2, \tag{14}$$

whose maximization for a given molecular wave function $|\Psi\rangle$ determines the sought-after orbital basis $\mathcal{B}_{MEAO}$ (see Fig. 1b). Here,

$$\Gamma_{k\sigma, lt}^{i\sigma, j\tau} = \langle \Psi | f_{i\sigma}^\dagger f_{j\tau}^\dagger f_{lt} f_{k\sigma} | \Psi \rangle, \tag{15}$$

are elements of the 2RDM $\Gamma$, and specifically $\Gamma_{j\uparrow, j\downarrow}^{i\uparrow, i\downarrow} = \langle 0, \uparrow\downarrow | \hat{\rho}_{ij} | \uparrow\downarrow, 0 \rangle$ and $\Gamma_{j\uparrow, i\downarrow}^{i\uparrow, j\downarrow} = \langle \uparrow, \downarrow | \hat{\rho}_{ij} | \downarrow, \uparrow \rangle$. The role of the proxy function $F_{MEAO}$ is not to replicate the exact orbital entanglement for all two-orbital reduced density matrices, but rather to ensure that its maximum identifies an orbital basis closely aligned with the one that maximizes the true entanglement, a behavior supported by the analytical evidence discussed above and further validated by the numerical results. In most cases where the bonding pattern is simple (e.g., typical Lewis structures), the wave function $|\Psi\rangle$ we use to compute the 2RDM (and correspondingly the objective function $F_{MEAO}$) can be a mean field wave function from a Hartree-Fock or a density functional theory calculation. In more challenging cases such as the dissociation of LiH, a correlated ground state (in our case a matrix product state, MPS) should be used.

Step 3: Grouping MEAOs into bonds. Bonds are identified by grouping MEAOs into strongly entangled pairs or clusters, which would require evaluating the entanglement between every inter-center pair of orbitals. This computationally expensive process can be efficiently approximated using the mutual information[29,41,79–82]:

$$I_{ij} \equiv I(\hat{\rho}_{ij}) = S(\hat{\rho}_i) + S(\hat{\rho}_j) - S(\hat{\rho}_{ij}), \tag{16}$$

which is computationally inexpensive and serves as an upper bound to the entanglement, $I_{ij} \geq E_{ij}$[41]. $I_{ij}$ is also referred to as the total correlation, or simply correlation, between the orbitals $i$ and $j$. To group MEAOs into bonds, we construct first a correlation graph: Two orbitals are considered connected in the correlation graph if their correlation $I_{ij}$ exceeds a threshold $\eta = 10\%$ of the maximum value $I_{max} = 2\log(4)$. Similar to the maximal value of entanglement, $I_{max}$ is also attained by the covalent bond state, Eq. (4). Introducing a threshold $\eta > 0$ reduces the number of orbitals to be analyzed by removing those that do not contribute meaningfully to bonding, while choosing $\eta$ small enough ensures all relevant orbitals are retained. The resulting clusters reliably correspond to two-center or multicenter bonds, as illustrated in Fig. 1c.

Step 4: Apply quantum information tools to each identified bond. Once the orbital pairs and clusters responsible for chemical bonding are identified, quantum information tools are applied to characterize each bond in greater detail. These include measures of bipartite and genuinely multipartite entanglement, which provide deeper insight into the underlying bonding structure. To this end, we employ correlated wave functions to achieve an accurate description of the ground state. Two-center bonds are characterized through the entanglement between the two orbitals forming a pair after the graph partition, whereas multicenter bonds are quantified by the GME among all orbitals in a bonding cluster containing more than two centers. To compute the entanglement between two MEAOs in a bond, we directly evaluate the matrix elements of the corresponding two-orbital reduced density matrix using an MPS ansatz. These matrix elements are components of particle reduced density matrices up to the fourth order. This approach remains computationally efficient, as only a small subset of these matrix elements is required due to symmetry constraints.

Due to the difficulty of computing bipartite entanglement for mixed states, the GME can be efficiently computed only for pure states. Therefore, to evaluate the GME, we employ a complete active space (CAS) wave function in which only the orbitals participating in the multicenter bond are included in the active space, while the remaining orbitals are transformed into closed and virtual orbitals as described in the following. We first partition the one-particle Hilbert space into an active subspace (comprising the multicenter bonding orbitals) and a non-active subspace (the remainder). Let $\gamma^{non-active}$ denote the restriction of the full spin-free one-particle reduced density matrix (1RDM), $\gamma_{ij} = \sum_{\sigma=\uparrow,\downarrow} \langle \Psi | f_{i\sigma}^\dagger f_{j\sigma} | \Psi \rangle$, to the non-active subspace. Since the wave function on the non-active orbitals corresponds to almost fully occupied local bonding orbitals and almost empty local antibonding orbitals, the eigenvalues of $\gamma^{non-active}$ are close to either 0 or 2. We assign the natural orbitals of $\gamma^{non-active}$ with eigenvalues greater than 1 as closed orbitals, and those with eigenvalues smaller than 1 as virtual orbitals. By construction, the closed and virtual orbitals are orthogonal to those involved in the multicenter bond, thereby defining a consistent active space partition.

Numerical details: The construction of intrinsic atomic orbitals and meta-Löwdin orbitals was performed using the PySCF package[83] with default settings. To find the maximally entangled atomic orbitals (MEAOs), the objective function, Eq. (14), was maximized using a second-order Newton-Raphson method, where the analytic gradient and diagonal elements of the Hessian were used, as detailed in Supplementary Note 1.

Molecular structures of the molecules in Table 1 and CO in Table 3 were taken from experimental data in the NIST Computational Chemistry Comparison and Benchmark Database[57], except for $He_2$ whose bond length is taken from ref. 84. The carbon rings in Table 2 and those in Fig. 5b-e are optimized using the B3LYP functional with the 6-311++G(d,p) basis[58]. In particular, the geometry optimization is performed with the PySCF package for $C_2H_5^+$, $C_3H_5^+$, $C_6H_8$, $C_6H_{10}$, and $C_6H_{12}$, while the optimized geometries of the rest of the carbon rings are directly taken from the Supplemental Material of ref. 58. The geometries of $Li_3^+$ and $Li_4$ are optimized geometries on CCSD and CCSDT levels of theory, respectively[85]. The structure of $Cr(CO)_6$ is constructed with the experimental bond lengths reported in ref. 86. All active space and full configuration interaction calculations were performed with an MPS and a density matrix renormalization group (DMRG) solver provided by Block2[87,88].

The relative entropy of bipartite entanglement for mixed states was calculated using the semidefinite programming package CVX[89,90] and quantum information helper functions provided in refs. 91,92. The algorithm for calculating the entanglement can be found in ref. 93.

The Diels-Alder reaction was simulated using the Gaussian program[94], starting from the transition state and progressing towards the reagents and products along separate paths. For the thermal state of LiH, the thermal 2RDM is used to optimize the MEAOs, starting from a set of meta-Löwdin localized orbitals. The thermal 2RDM is computed as the thermal average of the state-specific 2RDMs obtained via DMRG calculations correlating all electrons and all orbitals in the basis.

The minimization in the definition of GME is approximated by minimizing over all single orbital entropies, two orbital entropies, and bipartite entanglement along the MPS. This is a sensible approximation since the largest genuinely entanled cluster we have studied in this work consists of only six centers. In most cases, the minimization is realized by one of the single orbital entropy values. The aromatic indices in Fig. 5g were taken from ref. 58 and linearly renormalized to the interval [0,1], by first taking the absolute value, then subtracting by the lowest value, and finally divided by the largest value. Orbital isosurface plots were produced using the software Jmol[95].

## Data availability

The data that support the findings of this study are available from the corresponding authors upon request. Unprocessed energies and aromatic indices for the Diels-Alder reaction and molecular structures that are not listed in the NIST database are provided as Supplementary Data 1. Cube files for generating orbital isosurfaces are available from Zenodo[96]. Source data are provided with this paper.

## Code availability

The codes used in this study are available from Zenodo[96] and from the corresponding author upon request.

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

## Acknowledgements

We thank Jerzy Cioslowski and Stefan Knecht for helpful discussions on chemical bonding and Benjamin Yadin for insightful discussions on multipartite entanglement.

## Author contributions

The project was conceived by C.S. The theoretical framework was developed by L.D. with input from C.S., and the codes were implemented by L.D.; E.M. provided expertise in chemical bonding theory and aromaticity, designed most of the examples in the manuscript and contributed to the chemical interpretation of the results. The manuscript was drafted by L.D. and revised by all authors (L.D., E.M., and C.S.). Funding was acquired by C.S.

## Funding

We acknowledge financial support from the German Research Foundation (Grant SCHI 1476/1-1), the Munich Center for Quantum Science and Technology, and the Munich Quantum Valley, funded by the Bavarian state government through the Hightech Agenda Bayern Plus (L.D., C.S.). The grant PGC2018-098212-B-C21 funded by MCIN/AEI/10.13039/501100011033, "FEDER Una manera de hacer Europa", and the Basque Government (Project IT2067-26) are also acknowledged (E.M.). Open Access funding enabled and organized by Projekt DEAL.

## Competing interests

The authors declare no competing interests.
