## [Transparent Peer Review file · Nature Communications]

Chemical bonding concepts emerge naturally from maximally entangled atomic orbitals

Corresponding Author: Dr Christian Schilling

Version 0:

Reviewer comments:

Reviewer #1

(Remarks to the Author)

In this work, the authors try to extract information on chemical bonding from quantum information theory and entanglement measures. To this end, they perform a Hilbert space partitioning of atoms in molecules which comes, of course, with all associated limitations like the impact of very diffuse functions.

I am a bit surprised to see that no attention is paid to work that does something very similar without recourse to orbitals but using atoms in molecules theory and the QITAIM (Quantum Information Theory of Atoms in Molecules) published recently. Admittedly this is a different approach but one that does not rely on orbitals (Van Hende, D.; Van der Stichelen, R.; Bultinck, P.; Acke, G., Quantum Information Patterns between Atoms in a Molecule. Chem. Eur. J., 2024, 30, e20240081). In this paper, the case of H₂ is discussed, as well as some aromatic systems and the intricate case of biphenyl. It could be useful to compare some of these aspects and findings. For instance, delocalization indices can be zero whereas the entanglement can be very high (for example G.Acke, P.Bultinck, J. Mol. Model. 2018, 24.).

The paper, in my opinion, suffers from an apparent need to limit the theoretical derivations to a minimum and as a consequence, it leaves several aspects unclear. For instance, the so-called prototypical covalent bond is not clear to me. How was this chosen? Must I assume that the prototypical covalent bond corresponds to equation 1, say H₂ at the Hartree-Fock level? This is strange to me as a CI calculation, which has the deepest minimum on the PES, definitely does not correspond to equation 1.

In equation 3, Löwdin symmetrization is used, which depending on the case may significantly delocate the new basis functions from their original centers as the new basis functions are a linear combination of the old ones. This was in part a reason why the Hilbert path was abandoned in the paper cited above. Notation-wise, I have difficulty understanding the notations like L/R as a subscript. It is used in a different meaning above equation 3 (where it seems to mean to me L->+ and R->- to the right of the equality) and in equation 4 where it has a different meaning, I believe. At least, it could do with a tad more explanation as not everybody is familiar with the partial tracing which is obviously important here.

Inter-center entanglement measures are not explained but they play a key role. Why are these not properly introduced? What is the precise role of the CAS calculations? I do not understand the complete algorithm. It would seem as if a Hartree-Fock calculation is first done. Then there is the Hilbert space partitioning. Then MEAO. And then CAS calculations are performed. This is not explained in figure 1. And how are the QIT measures computed then? These do require the N-density matrix, no? or at what stage is another trace taken? Also, some discussions go past me. In table 1, it is claimed that pi bonds are generally weaker than sigma bonds, so what about I in acetylene? What precisely do I need to look at?

The paper does give a lot of results but I fear I cannot reproduce the data in the paper, as a referee should be able to do from the method details. Maybe the format of the journal is not appropriate for this, and so I would advise to either focus on a clearer explanation and less data or a more extensive paper in e.g., JCTC or first a theory paper elsewhere and then data in the current journal but now I fear the paper is quite opaque.

(Remarks on code availability)

I have found no reference to the code, like a repository.

Reviewer #2

(Remarks to the Author)

The work of Schilling and co-workers presents an overview to attempt to provide a unifying theory for chemical bonding. The results are too limited to meet this attempt successfully, because there is no hydrogen bonding no very sparse bonding such

as lithium clusters and no heavier elements.

The work appears to have been conducted carefully.

Your view on the potential significance of the conclusions for the field and related fields. If you think that other findings in the published literature compromise the manuscript's significance, please provide relevant references.

The conclusions are vague. A new theoretical approach is presented but the impact or benefits are not clear.

The results are well presented with clear figures.

The context is lacking in terms of going beyond "art for arts sake" what is the benefit of this approach?

(Remarks on code availability)

Reviewer #3

(Remarks to the Author)

The Maximally Entangled Atomic Orbitals (MEAOs) method provides a fresh approach to analyzing and understanding chemical bonding on the basis of quantum entanglement concepts. This, in of itself, is valuable. The paper is well written and well argued and the quantum chemical calculations are of a high quality. However, the method lacks in "predictive power", begging the question: What is the experimentally-verifiable prediction that this approach can deliver? Along these lines, what is the relation of MAEOs to experimental chemistry? This my major concern which has not been demonstrated in this study.

While MEAOs method does not explicitly predict new (or retrodict known) chemical phenomena, and to its credit, it does reveals and quantifies bonding features that may have been overlooked by traditional methods (QTAIM, ELF, etc.). For example, it seems capable of identifying bond types (which is in itself a somewhat arbitrary classification) without assuming predefined bonding models especially in the cases of multicenter bonds based on "entanglement strength" and does recover conventional bond orders for more common bonds. It shines best in describing the evolution of bonding along reaction paths particularly such as that of the Polanyi's harpoon mechanism (e.g. LiH dissociation) predicting a non-monotonic bond order along the IRC and also as a quantitative descriptor of aromaticity. It is not clear, however, if the method can also characterize weak bonding e.g. hydrogen bonding, van der Waals interactions, pi-stacking, etc.

In conclusion, the proposed method is insightful but not predictive. It is certainly worthy of publication in a specialized quantum chemistry journal.

(Remarks on code availability)

Version 1:

Reviewer comments:

Reviewer #1

(Remarks to the Author)

The authors have answered my questions in an appropriate way. They have also added details that make the paper more accessible.

I could now effectively reproduce some of the data reported in the paper.

(Remarks on code availability)

This repo contains some Python code with decent readme files. The users of the code are expected to have installed PySCF. It is in this code that most data are generated which are then post-processed via own codes on the github repo.

Reviewer #2

(Remarks to the Author)

The author has thoroughly addressed all my concerns. I recommend publication.

(Remarks on code availability)

The code was downloaded successfully from the given repository into a clean Python 3.12.3 virtual environment. The examples provided with the code appeared to execute without errors. The README file usefully specified the requirements, which were installed automatically as dependencies upon typing 'pip install'. Perhaps the recommendation to install in a clean virtual environment, and the command required to perform the installation, needs to be included in the README. The

former recommendation is important as the install required an older version of the 'numba','pyberny' and rather importantly 'numpy', modules than was currently installed in my main Python environment, potentially causing a version dependency clash with other installed software. The code is certainly useable 'as-is' for the community, with the small proviso that the file 'meao.py' and the example files needed minor edits (specifically 'import' statements for 'MAEO' and 'tools' modules) in order to run. The specific results of the paper were not cross-checked for reproducibility due to time constraints.

Reviewer #3

(Remarks to the Author)

The paper is improved, but I still think it lacks the general interest and urgency for a Nature Communications paper in particular since it does not predict any observable (experimentally verifiable/measurable) properties of the studied systems.

(Remarks on code availability)

Reply to the Referee

November 5, 2025

We thank the referees for their time, effort, and for the valuable and constructive comments. They acknowledged both the novelty and the potential of our work, and several positive aspects were explicitly noted, describing it as a ‘fresh approach’ that is ‘valuable,’ with a manuscript that is ‘well written’ and ‘well argued,’ and ‘quantum chemical calculations . . . of a high quality.’ This positive assessment aligns with the highly encouraging feedback we have received since submission, including invitations to present the MEAO framework at conferences and theoretical chemistry seminars, as well as active collaborations with leading developers to implement this potentially transformative bonding analysis framework into major software packages (OpenMolcas and DFTB+).

In the following, we reply to all points raised by the referees and explain how we have revised our manuscript accordingly. The text by the referees is presented in ‘green’, our responses in ‘black’, and the new text added to the manuscript in ‘blue’. To support the referees, we also provide a separate PDF file that highlights all changes.

Referee #1:

In this work, the authors try to extract information on chemical bonding from quantum information theory and entanglement measures. To this end, they perform a Hilbert space partition of atoms in molecules which comes, of course, with all associated limitations like the impact of very diffuse functions.

We note that robust Hilbert space partitions do not suffer from complications related to diffuse basis functions. This has been demonstrated for intrinsic atomic orbitals (IAOs), which form the basis of the maximally entangled atomic orbitals (MEAOs) introduced in this work, in one of the authors’ recent studies [10.1002/chem.202401282] (see Table 1).

Generally speaking, our theory indeed takes a Hilbert-space partition as its starting point, and the explicit derivation or optimization of such partitions is never the goal of our theory. Nevertheless, our framework is not disconnected from the partitioning problem. Namely, the effectiveness of the QIT-based bonding analysis produced from a given partition can retrospectively indicate that partition’s adequacy, so our results also serve as a practical diagnostic for comparing and assessing partitioning schemes.

1. I am a bit surprised to see that no attention is paid to work that does something very similar without recourse to orbitals but using atoms in molecules theory and the QITAIM (Quantum Information Theory of Atoms in Molecules) published recently. Admittedly this is a different approach but one that does not rely on orbitals (Van Hende, D.; Van der Stichelen, R.; Bultinck, P.; Acke, G., Quantum Information Patterns between Atoms in a Molecule. *Chem. Eur. J.*, 2024, 30, e20240081).

We have added the cited work and briefly discussed it in the Introduction (see below). While QITAIM provides an alternative real-space perspective, the authors of that work explicitly noted the difficulty of extracting entanglement entropies from real-space partitions using correlated wavefunctions, which restricted their analysis to mean-field (RHF/UHF) solutions. In contrast, we employ correlated wavefunctions ranging from CASCI to near-FCI (via DMRG), which is essential for accurate bonding analysis, particularly in challenging cases such as LiH dissociation.

To give a concrete example, when restricted to RHF, real-space and Hilbert-space partitions yield an identical entanglement value of $2 \ln(2)$ for H_2 at any bond length below the Coulson-Fischer point, as shown by the reference above and our previous study [L. Ding, Z. Zimborás, S. Knecht, C. Schilling, *Quantum Sci. Technol.* 8, 2023, Fig. 2(b,c)]. However, a clear distinction between the two approaches is that we are able to go beyond the limited RHF wavefunction, thereby recovering fractional bond order as the molecule dissociate.

The steep computational cost associated with manipulating a correlated wavefunction in a real-space partitioning arises because the number of electrons occupying an atomic subregion can be very large. Consequently, evaluating the reduced density matrix for such a region requires unfeasibly high orders of particle reduced density matrices. Our combination of Hilbert-space partitioning with MEAOs is free of such an issue. By representing two-center bonds as pairs of MEAOs, only selected elements of the two-, three-, and four-particle reduced density matrix are required. This comparison further highlights the computational and conceptual advantages of Hilbert-space partitioning for QIT-based bonding analysis.

“Due to these issues, a recent information theoretical approach to bonding analysis based on real-space partitioning, though conceptually interesting, has thus far been limited to mean-field descriptions only.[22]”

2. In this paper, the case of H_2 is discussed, as well as some aromatic systems and the intricate case of biphenyl. It could be useful to compare some of these aspects and findings. For instance, delocalization indices can be zero whereas the entanglement can be very high (for example G.Acke,P.Bultinck,J. Mol. Model. 2018,24.).

It is important to recall that entanglement is a *relative* concept, i.e., it depends on the choice of subsystems. When considered between localized orbitals, it generally correlates with electron delocalization. The paper cited by the referee shows that electron correlation (beyond HF) reduces the values of delocalization indices. However, this electron correlation must not be confused with orbital entanglement, as we have discussed in detail in our previous foundational works [L. Ding, C. Schilling, *J. Chem. Theory Comput.* 2020; L. Ding, S. Marzadad, *et al.*, *J. Chem. Theory Comput.* 2021]. The only type of orbital entanglement that can persist when electron delocalization is zero is spin–spin entanglement. Understanding this special case, and in particular its thermal non-robustness, represents an additional key contribution of our present work. We extended the concept introduced in [L. Ding, C. Schilling, *J. Chem. Theory Comput.* 2020] to the dissociation of LiH , thereby resolving the common issue of unphysical spin–spin entanglement in the dissociation limit by employing a thermal rather than a pure-state description. This achievement, which establishes a physically consistent and conceptually transparent treatment of spin entanglement at bond dissociation, represents a significant advance in its own right. It demonstrates the power of our systematic and comprehensive framework based on quantum information theory to uncover and resolve long-standing inconsistencies in bonding analysis.

3. The paper, in my opinion, suffers from an apparent need to limit the theoretical derivations to a minimum and as a consequence, it leaves several aspects unclear. For instance, the so-called prototypical covalent bond is not clear to me. How was this chosen? Must I assume that the prototypical covalent bond corresponds to equation 1, say H_2 at the Hartree-Fock level?

This is strange to me as a CI calculation, which has the deepest minimum on the PES, definitely does not correspond to equation 1.

The concept of orbital entanglement is well established in the literature, and numerous studies have introduced and applied this concept in different contexts. In our manuscript, we deliberately presented the key ideas at a high conceptual level to make the work accessible to a broader audience. We believe that combining established concepts from different fields in this manner is a particularly effective strategy for advancing scientific understanding. Nevertheless, following the referee’s suggestion, we have revised Section IIB to achieve a clearer balance between accessibility and technical detail.

To address the specific concern raised by the referee regarding the H₂ bond, we note that the RHF solution of the minimal H₂ model serves as the starting point for virtually all bonding theories, including the QITAIM reference mentioned by the referee. It is not only the simplest model but also one of the most meaningful, as it represents a rare case where the traditional integer bond is recovered within a quantum-mechanical framework. This is precisely why this particular state is considered the *prototype* of covalent bonding, from which typical multiple bonds can be systematically constructed.

On the one hand, the RHF solution of H₂ provides an essential consistency check for any bonding theory: in the absence of electron correlation, the predicted bond order should coincide with that from standard molecular orbital theory. On the other hand, the prototypical covalent bond state is, by construction, not the minimum on the potential energy surface, since the exact correlated ground state of H₂ is not a purely covalent bond. Thus, what the referee describes as a discrepancy in our treatment simply reflects the difference between two levels of theory, rather than any inconsistency in our formalism.

4. In equation 3, Löwdin symmetrization is used, which depending on the case may significantly delocate the new basis functions from their original centers as the new basis functions are a linear combination of the old ones. This was in part a reason why the Hilbert path was abandoned in the paper cited above.

In Eq. (3), we consider a two-orbital Hilbert space with D_{2h} symmetry. In this case, there exists only one possible way to localize the orbitals while preserving orthonormality, namely through Löwdin symmetrization. This is therefore not a deliberate methodological choice, but rather a direct consequence of the system’s symmetry and Hilbert space dimensionality.

5. Notation-wise, I have difficulty understanding the notations like L/R as a subscript. It is used in a different meaning above equation 3 (where it seems to mean to me L→ + and R→ – to the right of the equality) and in equation 4 where it has a different meaning, I believe. At least, it could do with a tad more explanation as not everybody is familiar with the partial tracing which is obviously important here.

The L/R notation is used to preserve the correspondence between the original non-orthogonal atomic orbitals $\varphi_{L/R}$ and the orthogonalized atomic orbitals $\tilde{\varphi}_{L/R}$. To avoid confusion, we modify the definition of the latter to one that does not refer anymore to the bonding/anti-bonding orbitals:

“To elaborate further on the idealized covalent bond and align our QIT perspective with its nonlocal character, we introduce the symmetrically orthogonalized atomic orbitals

$$\tilde{\varphi}_{L/R} = \frac{1}{2} \left(\frac{1}{\sqrt{1+S}} + \frac{1}{\sqrt{1-S}} \right) \varphi_{L/R} + \frac{1}{2} \left(\frac{1}{\sqrt{1+S}} - \frac{1}{\sqrt{1-S}} \right) \varphi_{R/L},$$

where $S = \langle \varphi_L | \varphi_R \rangle$.”

6. Inter-center entanglement measures are not explained but they play a key role. Why are these not properly introduced?

We have extended the original in-line definitions of entanglement and inter-center entanglement to standalone equations:

“Consequently, the closed formula (5) for entanglement in pure states is no longer applicable, and the entanglement between A and B must instead be determined numerically as the minimum “distance” of $\hat{\rho}_{AB}$ to the set of unentangled mixed states S_{sep} [40,41]

$$E(\hat{\rho}_{AB}) = \min_{\hat{\sigma} \in S_{\text{sep}}} \text{Tr}[\hat{\rho}_{AB}(\log \hat{\rho}_{AB} - \log \hat{\sigma})],$$

$$S_{\text{sep}} = \left\{ \sum_i p_i \hat{\sigma}_A^{(i)} \otimes \hat{\sigma}_B^{(i)}, p_i > 0, \sum_i p_i = 1 \right\}.$$

”

“ [...] MEAOs are a basis of localized orbitals $\mathcal{B}^{\text{MEAO}} = \{\{\phi_i^{(1)}\}, \{\phi_j^{(2)}\}, \dots, \{\phi_k^{(M)}\}\}$ that ideally maximizes the total inter-center entanglement, defined as the sum of entanglement between two orbitals sitting on different atoms

$$E_{\text{inter-center}} = \sum_{i < j} E(\hat{\rho}_{ij}), \quad (1)$$

where $\hat{\rho}_{ij}$ is the orbital RDM of orbital i and orbital j that sits on *different* atomic centers, obtained by tracing out all other orbital degrees of freedom from the overall wavefunction $|\Psi\rangle$. ”

7. What is the precise role of the CAS calculations? I do not understand the complete algorithm. It would seem as if a Hartree-Fock calculation is first done. Then there is the Hilbert space partition. Then MEAO. And then CAS calculations are performed. This is not explained in figure 1. And how are the QIT measures computed then? These do require the N-density matrix, no? or at what stage is another trace taken?

Based on the comments of all referees, we recognized the need to streamline the theoretical presentation to improve clarity and focus. We have therefore completely rewritten Section IIB to emphasize the key conceptual steps, while an extended description of the computational procedure is now provided in the Methods section. This restructuring avoids unnecessary technical detail in the main text and helps the reader focus on the central ideas. We refer the referee to the marked version of the revised manuscript for a detailed overview. For completeness, we reproduce below the relevant part of the extended Methods section that specifically addresses the role of the active space, the sequence of calculations, and the evaluation of the quantum

information measures based on the corresponding reduced density matrices.

“As discussed in the main text, the GME can be efficiently computed only for pure states. Therefore, to evaluate the GME, we employ a complete active space (CAS) wavefunction in which only the orbitals participating in the multicenter bond are included in the active space, while the remaining orbitals are transformed into closed and virtual orbitals in the following way. [...]”

“To compute the entanglement between two MEAOs in a bond, we directly evaluate the matrix elements of the corresponding two-orbital reduced density matrix using a matrix product state (MPS) ansatz. These matrix elements are components of particle reduced density matrices up to fourth order. This approach remains computationally efficient, as only a small subset of these matrix elements is required due to symmetry constraints. ”

8. Also, some discussions go past me. In table 1, it is claimed that pi bonds are generally weaker than sigma bonds, so what about I in acetylene? What precisely do I need to look at?

In acetylene, the mutual information associated with the π bond is indeed very similar to that of the σ bond. However, the entanglement (which represents the genuinely quantum part of the correlation) is slightly lower for the π bond. Consistent with this observation, in other systems featuring triple bonds, such as $\text{Cr}(\text{CO})_6$ and CO , we again find that the π bond exhibits less correlation and entanglement than the corresponding σ bond.

More generally, covalency does not necessarily equate to bond strength. Bonds characterized by smaller electron delocalization can still exhibit high rigidity and resistance to deformation. In the revised manuscript, we have therefore refined our wording and now use the more accurate terminology “lower/higher covalency” instead of “weaker/stronger.”

“The results also highlight chemical trends, such as π bonds being generally lower in covalency than σ bonds.”

9. The paper does give a lot of results but I fear I cannot reproduce the data in the paper, as a referee should be able to do from the method details. Maybe the format of the journal is not appropriate for this, and so I would advise to either focus on a clearer explanation and less data or a more extensive paper in e.g., JCTC or first a theory paper elsewhere and then data in the current journal but now I fear the paper is quite opaque.

The concept underlying the MEAO formalism is simple yet broadly applicable, allowing a wide scientific audience to appreciate its implications without the need for extensive numerical detail. While we recognize that researchers often specialize in distinct subfields of the quantum sciences, our work has greatly benefited from its interdisciplinary nature, bridging quantum information theory and quantum chemistry. For this reason, we consider *Nature Communications* an ideal platform for presenting this research at the interface of these two disciplines. The essential theoretical framework and results are fully conveyed in the main text, while the Supporting Information provides all technical details required for reproducibility.

To further enhance transparency and facilitate future applications, we have made our Python implementation of the MEAO formalism publicly available on GitHub [<https://github.com/schilling->

group/MEAO], together with illustrative examples demonstrating its use.

Referee #2:

1. The work of Schilling and co-workers presents an overview to attempt to provide a unifying theory for chemical bonding. The results are too limited to meet this attempt successfully, because there is no hydrogen bonding no very sparse bonding such as lithium clusters and no heavier elements. The work appears to have been conducted carefully.

We have revised both the title and the content of the manuscript to clarify that the MEAO framework provides a unifying theory for *covalent* bonding. In response to the comments by this referee and the third referee, we have substantially extended our analysis to a wide range of representative systems, including lithium clusters and the organometallic compound $\text{Cr}(\text{CO})_6$, as well as ionic (LiF) and weakly bound (He_2) molecules. The results are highly encouraging, in the sense that they confirm the robustness of the MEAO framework and its capacity to describe covalent bonding reliably across a broad spectrum of chemical systems. In particular, our measure of covalency faithfully reproduces the expected behavior: it quantitatively captures the covalent bond order where present, and correctly yields vanishing entanglement for systems such as LiF and He_2 , where no covalent bond exists. While the present MEAO framework, based solely on orbital entanglement, does not explicitly describe ionic or van der Waals interactions, it consistently distinguishes covalent from non-covalent bonding, providing a conceptually transparent and physically faithful measure of covalency. We are therefore grateful for the constructive comments from this and the third referee, which prompted us to broaden our analysis and ultimately strengthened the overall scope and clarity of the manuscript.

“Before we move on to more advanced applications of the MEAO formalism, we first point out that orbital entanglement directly captures only bonding effects arising from electron sharing and spin coupling, namely covalent bonding. Ionic or intermolecular bonding effects are only indirectly shown in the reduction of orbital entanglement. To demonstrate this constraint, we studied three molecules with increasing levels of ionicity, Li_2 , LiH , and LiF . As shown in the second panel in Table I, we indeed observe a consistent decrease going from Li_2 to LiF , in both mutual information and entanglement between the two most correlated MEAOs. Finally, we test against the van der Waals molecule He_2 , which was found to have zero correlation and entanglement between the MEAOs on the two helium atoms. We remark that this number of absolute zero is ensured by the fact that the minimal IAO subspace from which the He-MEAOs are constructed consists of only one orbital for each atom. And since both electrons are placed on this orbital, the total ground state is simply a product state between the two fully occupied He-MEAOs. To fully confirm the lack of covalency in He_2 , we extended the minimal IAO subspace to include the full basis set, finding only a vanishingly small amount of correlation between the two atomic centers.”

“To further validate the universal applicability of GME as a multicenter bond index beyond carbon rings, we also analyzed lithium clusters. While lithium dimers are described by standard two-center two-electron bonds, lithium trimer anions (Li_3^+) and tetramers Li_4 display clear multicenter bonding patterns, with GME value 0.908 and 0.905, respectively.”

“Our final demonstration of the MEAO framework involves chromium hexacarbonyl, $\text{Cr}(\text{CO})_6$,

a prototypical transition-metal complex. As an isolated molecule, CO exhibits a characteristic triple bond, comprising one σ and two π components. Upon coordination of six CO ligands to the Cr center, the C–O bond order decreases to a value between a double and a triple bond, in accordance with the well-known Dewar–Chatt–Duncanson mechanism of σ -donation and π -back-donation [71,72]. This reduction in covalency is also quantitatively shown by the diminished electron sharing between the C and O atoms in a subsequent study [73]. In Table III, we present the correlation and entanglement between the bonding MEAOs in both the free carbon monoxide and the chromium hexacarbonyl, correctly confirming the reduction in the covalent bond order between C and O. The MEAOs are optimized based on an IAO Hilbert space partition constructed from the entire atomic basis set (6-31G(d,p) basis set for C and O and Roos augmented double-zeta ANO for Cr). Correlation and entanglement between the MEAOs are evaluated from the one-particle reduced density matrix (1RDM) obtained in a B3LYP density-functional-theory calculation.”

2. [Your view on the potential significance of the conclusions for the field and related fields. If you think that other findings in the published literature compromise the manuscript’s significance, please provide relevant references.] The conclusions are vague. A new theoretical approach is presented but the impact or benefits are not clear. The results are well presented with clear figures. The context is lacking in terms of going beyond ”art for arts sake” what is the benefit of this approach?

Nowadays, computational chemists can simulate a vast range of systems, addressing problems that span chemistry, physics, and even biology. However, their role extends far beyond providing numerical values for observables that can often be measured experimentally. Their true added value lies in offering insight and rationalization. The empirical rules that physical scientists rely on to explain simple, classical structures often prove inadequate when tackling the most interesting and challenging problems.

For this reason, computational chemists are frequently consulted by colleagues seeking explanations of unusual electronic structures that defy traditional rules, for instance, molecules with atypical chemical bonds, highly symmetric rings suspected of being aromatic, pocket-like structures capable of holding an electron, or other features that cannot be described by classical bonding theories. Addressing such questions requires tools that can rationalize complex systems while, in limiting cases, recovering the familiar concepts of classical chemistry, thereby bridging the gap between classical models and real molecular systems.

It is therefore difficult to overstate the importance of developing a theory of covalent bonding that is firmly rooted in physical principles and capable of explaining not only fundamental molecules, but also transition-metal complexes, bond formation along electrocyclic reaction pathways, and the loss of aromaticity under structural distortion. Our MEAO framework achieves exactly this. It provides, for the first time, a unified and quantitative description of covalent bonding derived from quantum information theory, able to recover classical bonding concepts in simple systems while extending seamlessly to complex molecular environments. We believe that this development represents a major conceptual advance and that the MEAO framework is poised to become an essential component of the computational chemist’s toolkit.

Referee #3:

The Maximally Entangled Atomic Orbitals (MEAOs) method provides a fresh approach to analyzing and understanding chemical bonding on the basis of quantum entanglement concepts. This, in of itself, is valuable. The paper is well written and well argued and the quantum chemical calculations are of a high quality.

1. However, the method lacks in “predictive power”, begging the question: What is the experimentally-verifiable prediction that this approach can deliver? Along these lines, what is the relation of MAEOs to experimental chemistry? This my major concern which has not been demonstrated in this study.

The MEAO framework provides predictive power in a conceptual sense: it enables the quantitative identification and characterization of covalent bonding directly from the electronic wavefunction, without relying on empirical input or predefined bonding models. This predictive capability manifests in its ability to recover established bonding patterns in simple molecules and to rationalize complex cases — such as transition-metal complexes, reaction pathways, and aromatic systems — on the same rigorous footing. While MEAOs do not predict experimental observables directly, they yield transferable and physically grounded quantities, such as covalent bond orders and measures of delocalization, that can be directly correlated with experimentally accessible properties like bond energies, vibrational frequencies, or spectroscopic signatures. In this sense, the MEAO framework provides the essential theoretical link between quantum mechanical structure and experimentally observed chemical behavior.

2. While MEAOs method does not explicitly predict new (or retrodict known) chemical phenomena, and to its credit, it does reveals and quantifies bonding features that may have been overlooked by traditional methods (QTAIM, ELF, etc.). For example, it seems capable of identifying bond types (which is in itself a somewhat arbitrary classification) without assuming predefined bonding models especially in the cases of multicenter bonds based on “entanglement strength” and does recover conventional bond orders for more common bonds. It shines best in describing the evolution of bonding along reaction paths particularly such as that of the Polanyi’s harpoon mechanism (e.g. LiH dissociation) predicting a non-monotonic bond order along the IRC and also as a quantitative descriptor of aromaticity. It is not clear, however, if the method can also characterize weak bonding e.g. hydrogen bonding, van der Waals interactions, pi-stacking, etc.

As Referee 3 pointed out, challenging cases of covalent bonding are indeed where the MEAO framework shines best. We have therefore refined both the title and the scope of the paper to focus explicitly on covalent bonding. To establish clear limits to this scope, we applied the MEAO framework to weakly bound molecules such as He₂ and Ar₂, for which the entanglement correctly approaches zero. This confirms that MEAO faithfully reproduces the absence of covalent bonding in such systems and does not directly characterize weak interactions. Furthermore, the observed decrease in entanglement from Li₂ to LiH and then to LiF shows that MEAO reliably detects the reduction in covalency while not attempting to quantify ionicity. These additional results not only clarify the boundaries of applicability of the formalism but also demonstrate its robustness and internal consistency as a physically grounded measure of covalent bonding.

“Before we move on to more advanced applications of the MEAO formalism, we first point

out that orbital entanglement directly captures only bonding effects arising from electron sharing and spin coupling, namely covalent bonding. Ionic or intermolecular bonding effects are only indirectly shown in the reduction of orbital entanglement. To demonstrate this constraint, we studied three molecules with increasing levels of ionicity, Li_2 , LiH , and LiF . As shown in the second panel in Table I, we indeed observe a consistent decrease going from Li_2 to LiF , in both mutual information and entanglement between the two most correlated MEAOs. Finally, we test against the van der Waals molecule He_2 , which was found to have zero correlation and entanglement between the MEAOs on the two helium atoms. We remark that this number of absolute zero is ensured by the fact that the minimal IAO subspace from which the He-MEAOs are constructed consists of only one orbital for each atom. And since both electrons are placed on this orbital, the total ground state is simply a product state between the two fully occupied He-MEAOs. To fully confirm the lack of covalency in He_2 , we extended the minimal IAO subspace to include the full basis set, finding only a vanishingly small amount of correlation between the two atomic centers.”

In conclusion, the proposed method is insightful but not predictive. It is certainly worthy of publication in a specialized quantum chemistry journal.

The ability to move beyond classical bonding rules while retaining their intuitive appeal is central to modern computational chemistry. Our physically grounded theory of covalent bonding, capable of describing systems ranging from simple molecules to transition-metal complexes and aromatic distortions, provides exactly this bridge. The MEAO framework combines rigorous quantum information principles with the interpretability of traditional bonding theory, offering a unifying and conceptually novel perspective on chemical bonding. Its relevance clearly extends beyond the boundaries of quantum chemistry, as it addresses a fundamental concept of chemistry through a framework grounded in first principles that also resonates with the broader physics and materials science communities. For this reason, the work would not be appropriately confined to a specialized quantum chemistry journal, where the emphasis is typically on methodological or computational details. Instead, its interdisciplinary scope, conceptual depth, and wide applicability make it ideally suited for the broad readership of *Nature Communications*.

Concluding Remarks

In summary, the revisions prompted by the referees’ constructive comments have significantly strengthened both the content and presentation of our work. The feedback on the scope and applicability of the MEAO framework led us to refine its focus and broaden the range of systems analyzed, firmly establishing its role as a unifying theory of covalent bonding. We have also clarified the conceptual foundations and computational workflow, thereby addressing the main concerns regarding scope, predictive power, and clarity. We are grateful to the referees for their valuable input and believe that, in its present form, the paper meets the standards and expectations of *Nature Communications*, combining conceptual novelty, broad relevance, and clear presentation.

Reply to the Reviewers

May 8, 2026

Reviewer #1:

Remarks to the Author: The authors have answered my questions in an appropriate way. They have also added details that make the paper more accessible. I could now effectively reproduce some of the data reported in the paper.

We are glad to hear that the reviewer found our manuscript improved and more accessible.

Remarks on code availability: This repo contains some Python code with decent readme files. The users of the code are expected to have installed PySCF. It is in this code that most data are generated which are then post-processed via own codes on the github repo.

We are pleased to see that our results can be reproduced using the codes in the Github repository.

Reviewer #2:

Remarks to the Author: The author has thoroughly addressed all my concerns. I recommend publication.

We thank the reviewer for the support of the publication.

Remarks on code availability: The code was downloaded successfully from the given repository into a clean Python 3.12.3 virtual environment. The examples provided with the code appeared to execute without errors. The README file usefully specified the requirements, which were installed automatically as dependencies upon typing 'pip install'. Perhaps the recommendation to install in a clean virtual environment, and the command required to perform the installation, needs to be included in the README. The former recommendation is important as the install required an older version of the 'numba', 'pyberny' and rather importantly 'numpy', modules than was currently installed in my main Python environment, potentially causing a version dependency clash with other installed software. The code is certainly useable 'as-is' for the community, with the small proviso that the file 'meao.py' and the example files needed minor edits (specifically 'import' statements for 'MAEO' and 'tools' modules) in order to run. The specific results of the paper were not cross-checked for reproducibility due to time constraints.

We thank the reviewer for the suggestions. The use of a virtual python environment is now recommended in the README file on the Github page. We also fixed the import issue. A list of requirements for the virtual environment is now provided, with version details for the required packages.

Reviewer #3:

Remarks to the Author: The paper is improved, but I still think it lacks the general interest and urgency for a Nature Communications paper in particular since it does not predict any observable (experimentally verifiable/measurable) properties of the studied systems.

We are pleased to hear that the paper is improved after revision. However, we do not agree with the statement that our entanglement-based bonding analysis does not provide experimentally verifiable predictions. Although chemical bonds are conceptual constructs that help chemists understand molecules and materials, the consequences of these understandings are highly observable and quantifiable. For example, the assignment of bond orders or multicenter bonding structures predicts the properties and reactivity of molecules. Aromaticity, which is measured by the genuine multipartite entanglement in this work, is responsible for an induced ring current when the molecule is placed under a magnetic field. We therefore argue that our framework is not only a new perspective on bonding analysis for chemical compounds but also a powerful tool for chemically relevant experimental predictions.